# Preparation of Lateral Flow PVDF Membrane via Combined Vapor- and Non-Solvent-Induced Phase Separation (V-NIPS)

**DOI:** 10.3390/membranes13010091

**Published:** 2023-01-10

**Authors:** Xiaoyun Wang, Dejian Chen, Ting He, Yue Zhou, Li Tian, Zhaohui Wang, Zhaoliang Cui

**Affiliations:** 1State Key Laboratory of Materials-Oriented Chemical Engineering, College of Chemical Engineering, Nanjing Tech University, Nanjing 210009, China; 2National Engineering Research Center for Special Separation Membrane, Nanjing Tech University, Nanjing 210009, China; 3Jiangsu National Synergetic Innovation Center for Advanced Materials (SICAM), Nanjing Tech University, Nanjing 210009, China; 4Nanjing Jiuying Membrane Technologies Co., Ltd., Nanjing 211899, China

**Keywords:** Poly(vinylidene fluoride), vapor non-solvent-induced phase separation, detection of colloidal gold, capillary flow rate

## Abstract

A large pore size Poly(vinylidene fluoride) (PVDF) membrane was prepared by the V-NIPS method using PVDF/N, N-dimethylacetamide (DMAc)/Polyvinyl pyrrolidone (PVP)/Polyethylene glycol (PEG) system. Firstly, the effect of different additive ratios on the membrane morphology and pore size was studied, and it was found that when the PVP:PEG ratio was 8:2, PVDF membranes with a relatively large pore size tend to be formed; the pore size is about 7.5 µm. Then, the effects of different exposure time on the membrane morphology and pore size were investigated, and it was found that as the vapor temperature increased, the pores on the surface of the membrane first became slightly smaller and then increased. Finally, the effects of different vapor temperatures on the membrane properties were discussed. The results showed that the as-prepared membrane exhibited suitable capillary flow rate and similar performance compared with a commercially available membrane in colloidal gold tests. The likely cause is that the amount of negative charge is less and the capillary migration rate is too fast. This paper provides a reference for the preparation of PVDF colloidal gold detection membrane.

## 1. Introduction

Lateral flow assays (LFAs) first emerged in the late 1960s, for monitoring serum proteins. In 1976, LFA was first used to detect human chronic gonadotropin (HCG) in urine [1]. From that time on, it has been widely used for rapid detection of biomarkers of infections and various diseases, such as human immunodeficiency virus (HIV), malaria and dengue virus [2]. As shown in Figure 1, the lateral flow test strip is composed of several parts: a sample pad, a conjugate pad, a reaction membrane (usually nitrocellulose membrane) and an absorbent pad, of which nitrocellulose (NC) membrane is the most critical material.

NC membranes used for lateral flow testing require the two most relevant parameters: (1) protein-binding capacity, and (2) protein capillary flow rate along the long axis [3]. Firstly, the strong dipole of nitrate interacts with the peptide bond of the antibody, which immobilizes the antibody on the NC membrane through electrostatic interaction; usually, these peptide bond cannot be fixed directly on NC membrane due to their small molecular weight. Therefore, small molecules are often chemically coupled to large molecules such as bovine serum albumin (BSA) and then fixed on NC membrane. BSA was used as the basis to evaluate the adsorption capacity of protein. Secondly, the NC membrane itself is a hydrophobic material, but a surfactant is added during its membrane formation, and the prepared membrane has a large pore size, so it can ensure that the sample to be tested moves along the NC membrane through capillary action [1]. However, NC membrane is a dangerous chemical and is extremely flammable, so special attention should be paid during transportation and storage. Moreover, with the development of LFA, the mechanical strength and protein adsorption capacity of NC membrane can no longer meet the current application requirements. Therefore, it is necessary to develop new membrane materials for LFA.

Poly(vinylidene fluoride) (PVDF) is a semi-crystalline polymer with -(CH_2_-CF_2_)_n_- repeating units. It presents high thermal, chemical and aging resistance [4,5]. PVDF membranes have been used in many application fields, including ultrafiltration (UF), microfiltration (MF), membrane bioreactor (MBR), membrane distillation (MD) and so on [4]. In addition, PVDF membranes also have some applications in biology. Ideris et al. reported the effect of solvent dissolution temperature in the casting solution on PVDF membrane protein binding. When the dissolution temperature was below 40 °C, the crystal structure of PVDF had a greater influence on its protein-binding ability, and the lower the *α* form, the stronger the protein-binding ability. However, above 40 °C, protein binding was more affected by porosity, with more protein retained at higher porosity [6]. Furthermore, additives are also important factors affecting membrane structure and properties. The most common polymer additives are hydrophilic polymers, such as polyvinylpyrrolidone (PVP) and polyethylene glycol (PEG). In the film-forming process, hydrophilic additives can promote the formation of pores, improve the connectivity of pores, increase the water flux of the membrane, and improve the hydrophilicity and pollution resistance of the membrane. Li et al. studied the effect of PVP as an additive on the structure and properties of PVDF hollow fiber membranes, and found that when using lower molecular weight PVP, the permeability and retention performance of the membrane were improved, whereas the higher molecular weight PVP would make the membrane structure dense, thereby reducing its permeability [7]. Lin et al. found that when polyethylene glycol (PEG) was used as an additive to make PVDF membrane, the surface pore size of the membrane increased with the increase in PEG molecular weight, and the flux and retention rate were the opposite, but increased with the increase in PEG content [8]. In this paper, PVP and PEG were selected as additives, and their molecular weight, concentration and proportion parameters were adjusted to improve the flatness, pore size uniformity and capillary migration rate of the film. However, there is no literature on the preparation of large-pore-size PVDF membranes and their application in LFA. Therefore, this paper discusses the conditions for the preparation of large-pore-size PVDF membranes, such as the ratio of additives, exposure time and temperature, etc. In addition, this paper used the prepared membrane in LFA.

Non-solvent-induced phase separation (NIPS) is a common method for preparing PVDF membranes. It involves pouring a polymer solution onto a suitable carrier and then immersing it in a coagulation bath containing a non-solvent. The exchange of the solvent in the polymer solution with the non-solvent from the coagulation bath leads to phase separation. Vapor-induced phase separation (VIPS) is a process in which the wet membrane is first exposed to a gaseous non-solvent at a given temperature and humidity for a period of time, and then immersed in a non-solvent solidification bath. If exposure time is moderate, the phase transformation mechanism will be a combination of VIPS and NIPS (i.e., N-VIPS).

## 2. Experimental Procedure

### 2.1. Material

PVDF 6010 (Mw = 600,000) powder was purchased from Shanghai Solef Co., Ltd., Shanghai, China. Dimethylacetamide (DMAc) (>99.8%) used as solvent was purchased from Shanghai Aladdin Reagent Co., Ltd., Shanghai, China. Polyvinylpyrrolidone (PVP K90, Mw = 360,000) was purchased from TCI Co., Ltd. Shanghai, China. Polyethylene glycol (PEG 600) was purchased from Beijing OKA Biological Technology Co., Ltd., Beijing, China. Bovine albumin (BSA, A1933-5G) was purchased from Anhui Senrise Technology Co., Ltd. Fuyang, China.

### 2.2. PVDF Membrane Preparation

A quantity of PVDF 6010 and additives (PVP K90 and PEG 600) was dissolved in DMAc in a flask, with the temperature set at 60 °C to promote full dissolution, and stirred for 12 h, in order to obtain a homogeneous casting solution. The mass fraction of PVDF was 16%, the mass fraction of DMAc was 74% and the total mass fraction of additives was 10%. The solution was then degassed until it became clear of bubbles. A desktop coater (with the knife height adjusted to 250-μm and speed set to 1.0 m∙min^−1^) was used to scrape the membrane. Both glass and nascent membrane were then placed on a water bath at 90% humidity at 60 °C and exposed for a while. The exposure was aimed at exchanging the solvent with water vapor. Then the glass was placed in ambient deionized water (DI water) for further solvent exchange. Finally, the resulting membranes were washed several times with DI water to remove residual DMAc, and then dried in air. The detailed process is shown in Figure 2 and Table 1.

### 2.3. PVDF Membrane Characterization

The surface and cross-sectional morphologies of the prepared PVDF dry membranes were observed by cold field emission scanning electron microscope (FESEM, S-4800, Hitachi Limited, Tokyo, Japan). All samples were freeze-fractured in liquid nitrogen and sputtered with gold membranes.

A X-ray diffractometer (XRD, Miniflex 600, Rigaku Corporation, Tokyo, Japan) with a Cu target was used to determine the crystallinity of the prepared PVDF membranes. Operating parameters included tube current and acceleration voltage of 15 mA and 40 kV, respectively; angle range of 5–60°; and scan speed of 5°∙min^−1^.

The functional groups in the prepared membranes were analyzed using a Fourier Transform infrared spectrometer (FTIR, Nicolet 8700, Thermo Fisher Scientific, Waltham, MA, USA). In this section, a diaphragm of a certain size was placed on the sample rack and scanned 32 times in attenuated total reflection (ATR) mode with a resolution of 4 cm^−1^ and a test wave number ranging from 500 to 4000 cm^−1^

An electrokinetic analyzer (SurPASS 3, Anton-Paar, Graz, Austria) was used to determine the surface charge of the membranes. Prior to analysis, the membrane was air-dried and the machine was cleaned with DI water. Then, 0.015 g potassium chloride (KCl) was dissolved in 250 mL deionized water to prepare KCl electrolyte solution. The streaming potential was determined by using 0.1 M HCl and NaOH to adjust the pH.

The porosity of PVDF membrane was measured by the gravimetric method. The sample was soaked in kerosene for 24 h until the membrane was completely wetted, and then the kerosene on the surface of the membrane was wiped away, leaving only the kerosene remaining in the membrane holes. The weight of the dry membrane and the wet membrane were recorded as *m*_0_ and *m*_1_, respectively. The following formula was used to calculate the porosity of the membrane:(1)$$\varepsilon =\frac{\frac{m_{1}-m_{0}}{\rho_{k}}}{\frac{m_{1}-m_{0}}{\rho_{k}}+\frac{m_{0}}{\rho_{p}}}\times 100\%$$

Here, *ρ*_k_ and *ρ*_p_ are the density of kerosene (about 0.81 g/cm^3^) and PVDF (about 1.78 g/cm^3^), respectively.

A tensile testing instrument (HLD 1000, Handpi, Jinhua, China) was used to measure the mechanical properties of prepared membranes. Each sample was first cut into 5 cm long and 1 cm wide strips using a Japanese knife mold. The thickness of each sample was measured by an electronic digital membrane thickness meter before the test. Both ends of the sample were fixed and stretched at a constant rate of 5 mm/min (25 °C). The electronic tension meter was set to zero and the original length of the sample *L*_0_ (cm) was recorded. It was slowly stretch until the test sample broke. The sample length *L* (cm) at this time and the tension data *F* (N) shown by the tension meter were recorded. The formula for calculating the elongation at break and tensile strength of the membrane are as follows:(2)$$\theta =\frac{L-L_{0}}{L_{0}}\times 100\%$$
(3)$$\sigma =\frac{F}{A}$$

The pore size of prepared PVDF membranes was determined using a Liquid–Liquid Pore Size Distribution Meter (GaoQ-PSMA-10, GaoQ, Nanjing, China).

The capillary migration rate of the prepared membrane was measured by the following steps. The sample was cut into 2 cm wide by 6 cm long pieces and DI water was used as the migration medium. Experiments were performed at room temperature (27 °C) and ambient pressure. The measurement started when the membrane was in contact with deionized water and the water migrated along the membrane surface for about 5 mm, and stopped when the water migrated to the membrane surface for 4 cm. The experimental process is shown in Figure 3.

The membrane-binding ability was evaluated by a model protein BSA. Each sample was cut into 1 cm wide by 2 cm long membrane strips and put into a conical beaker containing 4 mL of BSA solution (1 mg/mL). The mixture was incubated at 37 °C for 5 h with a shaking speed of 100 rpm. Each membrane was measured three times and averaged.

## 3. Results and Discussion

### 3.1. Additive Ratios

The morphologies of the membranes prepared with different additive ratios are shown in Figure 4a. The morphologies of membranes M1 and M2 are relatively similar, with a sponge-like structure in cross section, whereas M3 has a large micron-scale cavity in the cross section. This is because when the PVP content is low, the liquid–liquid phase separation occurs mainly in the casting liquid, and it is easier to form spongy pores [9,10]. It is also because the addition of PEG will increase the flatness of the membrane and the uniformity of the membrane. As shown in Table 2, with the increase in PVP content, the viscosity of the casting liquid increases, which slows down the exchange of solvent and non-solvent [11]. At this time, PVP is a high-molecular-weight additive, which increases viscosity, and delays the phase separation. After partially aggregated PVP is eluted, a large cavity shape appears [12,13].

The pore size distributions of membranes prepared with different additive ratios are shown in Figure 4b. When the ratio of PVP to PEG is 4:6 and 6:4, the pore sizes of the prepared membranes are about 1.3 microns and 1.2 microns, respectively; when the ratio reaches 8:2, the pore size of the prepared membrane is about 7 microns, which is consistent with the SEM photos. This is because when the ratio of PVP and PEG is relatively close, the viscosity of the casting solution is not much different, but when the ratio of PVP and PEG reaches 8:2, the viscosity of the casting solution increases significantly, hindering the phase separation kinetics. The hydrophilicity significantly improves the thermodynamics of phase separation. For the polymer-solvent-non-solvent ternary system, hydrophilic PVP with larger molecular weight can replace part of the polymer and increase the miscible region; the affinity with polymers and solvents is changed, increasing the thermodynamic in-stability of the casting film; and the binodal curve becomes closer to the (polymer)–(solvent/non-solvent) axis. PEG with smaller molecular weight can be regarded as a weak non-solvent. When replacing part of the solvent in the casting solution, the difference in the composition of different parts of the casting solution caused by the diffusion rate in the film-forming process can be reduced. The effect is to reduce the time difference in the phase-separation process and improve the uniformity of the membrane aperture.

The combined effect of PVP K90 on thermodynamic and kinetic phase-separation mechanisms can explain the significant increase in membrane pore size. Due to the expansion of the molecular chain, the fluidity of the PVP K90 chain is limited, so the outflow rate of PVP in the solidification bath is slower. However, the presence of a large amount of PVP in the solution increases the phase conversion rate, resulting in a large pore size. In addition, some of the retained molecules of PVP K90 leave the primary membrane and dissolve in water after delamination, forming further pores within the polymer membrane. In addition, the significant increase in pore size is due to the slow rate of non-solvent exchange during the VIPS process, and nucleation plays a crucial role in the dilute phase of the polymer, contributing to the formation of large-pore-size membranes [14].

### 3.2. Exposure Time

The morphologies of membranes prepared with different exposure times are shown in Figure 5a. When the exposure time was 15 s, the membrane showed a finger-like structure with a dense layer on top. Due to the short exposure time, the casting liquid quickly entered the NIPS process. Rapid exchange occurs because of a high affinity between non-solvent and solvent. Therefore, the solvent flowing out into the coagulation bath brought the polymer chains to the top surface of the membrane, resulting in a higher polymer concentration on the top surface of the membrane, forming a dense skin layer [15]. When the exposure time was 30 s, pores began to appear on the surface, finger-like pores still appeared on the cross section, and large cavities began to appear. When the exposure time was 1 min, macropores began to appear on the surface of the membrane, and the cross-section showed a micron-scale cavity structure. With the increase in exposure time, the number of pores on the surface of the membrane increased, and the exposure time of 3–10 min had little effect on the cross section, and all showed micron-scale cavity structure. This is because as the exposure time increases, especially when the exposure time is greater than three minutes, the casting film solution enters the NIPS process from the previous state very quickly, and thus it first goes through a longer VIPS process and then enters the NIPS process; therefore, the solvent and non-solvent exchange rates slow down, and pores begin to appear on the membrane surface. The cross-sectional structure changes from a finger-like macroporous structure to a sponge-like structure [16,17]. However, due to the high molecular weight of PVP, agglomeration occurs with the increase in VIPS time, so a large amount of PVP is eluted during the NIPS process, resulting in large cavities in the cross-sectional structure.

The effect of exposure time on the membrane pore size distribution is shown in Figure 5b, which also includes the morphologies of the membranes after drying. It can be seen from the figure that the pore size distribution of the membranes prepared with different exposure times is narrow, and from the morphologies of the membranes after drying, it can be seen that the membranes prepared with the exposure time of 15 s^−1^min are all curled. The reason is that when the exposure time is short, the PVP in the casting solution does not aggregate and may diffuse out in a large amount during the phase separation process, whereas with the increase in the exposure time, the PVP aggregates and participates in the membrane formation. Because PVP is a hydrophilic additive [18], the membrane flatness is high after drying. The change of exposure time not only changes the composition of the casting solution, but also changes the aggregated structure of the polymer in the casting solution. The influence of humidity is obvious. Chen et al. found that at low humidity, the membrane aperture changed little with the increase in exposure time, whereas at high humidity, the membrane aperture increased with the increase in exposure time [19]. Dehban et al. reported that in the indoor environment, the membrane aperture showed a trend of first increasing and then decreasing and then increasing with the increase in exposure time [20]. Therefore, there is no specific relationship between exposure time and aperture and the effect of exposure time on the pore size of the membrane is unclear.

### 3.3. Vapor Temperature

The morphologies of membranes prepared at different vapor temperatures are shown in Figure 6. The cross sections of membranes prepared at different vapor temperatures are not very different, and all of them are partially sponge-like and contain larger micron-scale cavities, but the surface of the prepared membrane is slightly different. As the vapor temperature increases, the pores on the surface of the membrane first become slightly smaller and then increase. This may be because when the solvent evaporation temperature is lower than the dissolution temperature, the mass transfer rate of the VIPS process is slow, so that a large amount of PVP is eluted on the membrane surface. When the evaporation temperature is higher than the dissolution temperature, the phase-transition mechanism is mainly the combination of VIPS and NIPS, which also leads to the elution of a large amount of PVP.

Figure 7a shows the pore size distribution of PVDF membranes prepared at different vapor temperatures. It can be seen from the figure that the pore size distribution of the membrane prepared at the vapor temperature of 60 °C is the most concentrated, around 8 microns. In addition, the membranes prepared at 50 °C and 70 °C are concentrated at around 16 and 20 microns, respectively. This is consistent with the SEM photo results.

Figure 7b shows the result of XRD study of the prepared membranes. The crystallites of the membranes prepared at different vapor temperatures are a mixture of *α* and *β* forms. The XRD patterns of the prepared membranes show that the PVDF crystallites have three characteristic peaks at ca. 18.3° (*α*), 20.1° (*β*), and 26.56° (*α*). Polarized FTIR spectra (Figure 7c) were used to characterize the crystalline characteristic peaks of the PVDF membranes. The characteristic peaks appear at 762, 796, 876, 1070, 1178 and 1423 cm^−1^, which are similar to those observed on PVDF membranes mainly containing *α* phase, whereas other crystal characteristic peaks at 840 and 1402 cm^−1^ can be attributed to the *β* phase [21,22].

Figure 7d shows the zeta potentials of PVDF membranes prepared at different vapor temperatures. At pH = 7, the surface of the membrane prepared at 60 °C had the greatest negative charge. Since the pore size of the membrane prepared at 60 °C is relatively small, it indicates that more PVP was involved in the membrane formation on the membrane surface. PVP is a hydrophilic material [15], which reduces the absorption of negative ions on the membrane surface in the electrolyte solution, thereby reducing the negative zeta potential of the membrane surface [23].

The capillary flow time and porosity of membranes prepared at different vapor temperature are shown in Figure 7e. With the increase in the vapor temperature, the capillary flow time shows a gradually decreasing trend. The porosity of several membranes is not very different, but due to the increase in temperature, the mass-transfer rate increases [24], and the exchange rate of solvent and non-solvent increases during the VIPS process, which improves the internal connectivity of the membrane, thereby decreasing its capillary flow time.

The tensile strength and elongation at break of membranes prepared with different vapor temperatures are shown in Figure 7f. As the vapor temperature increases, the tensile strength shows a slightly increasing trend, which is similar to the findings of Zhao et al. [25]. This may be because the formation of spherulites in the PVDF membrane is inhibited with increasing vapor temperature, resulting in increased membrane strength. The elongation at break shows a trend of first increasing and then decreasing.

Figure 7g shows the protein-binding capacity of PVDF membranes prepared at different vapor temperatures. It can be seen from the figure that there is little difference in the protein adsorption capacity of several membranes, at about 15 µg/cm^2^. The protein adsorption capacity of the membrane is related to many factors, such as pore size, porosity and crystal form [6]. The protein adsorption capacity of these membranes is low and close, because the porosity of the membranes is similar and the pore size is larger, and the protein cannot be fixed in the pores of the membranes.

Figure 8 shows the application of PVDF membranes prepared at different vapor temperatures in the actual colloidal gold test strips. As can be seen from the figure, compared with the commercially available NC membrane, the test line T of the self-made membrane did not develop color, and the quality-control line C developed a light color. This may be because the protein adsorption capacity of these membranes was low, and the surface had less negative charge, and could not form stable binding with the antibodies immobilized on the surface. The capillary flow rate is too fast, making it difficult for unadsorbed antibodies to be firmly anchored on the membrane surface. It is easily washed away by the sample to be tested, resulting in an inconspicuous line display [3].

## 4. Conclusions

In this paper, we proposed a method to prepare large-pore-size PVDF membranes, and use them for immunoassay. The morphologies of the membranes prepared at different additive ratios, exposure times and vapor temperatures were explored. The results showed that when the ratio of PVP to PEG was 8:2, the pore size of the membrane was the largest, because the high-molecular-weight additives of PVP agglomerated and formed larger cavities after elution. As the exposure time increased from 10 s to 1 min, the membrane surface gradually changed from dense to porous because solvent and non-solvent exchange became slower; however, the effect of exposure time on the pore size of the membrane is unclear. Membrane morphology and structure are similar at different vapor temperatures, and they exist as a mixture of *α* and *β* forms. When the vapor temperature was 60 °C, the membrane had the most concentrated pore size distribution and the greatest negative charge. As the vapor temperature increased, some other properties also changed. The capillary flow rate decreased and the strength and elongation at break increased. Finally, the PVDF membrane we prepared has a certain capillary flow rate and protein adsorption performance, which proves to be effective for colloidal gold tests. The protein adsorption capacity of the membrane can be improved by surface modification, and the influence of humidity should be further investigated. In addition, the surface flatness of the self-made membrane is lower than that of the commercial membrane. Therefore, it is necessary to optimize the formula and improve the overall uniformity of the PVDF membrane.

## Figures and Tables

**Figure 1 membranes-13-00091-f001:**
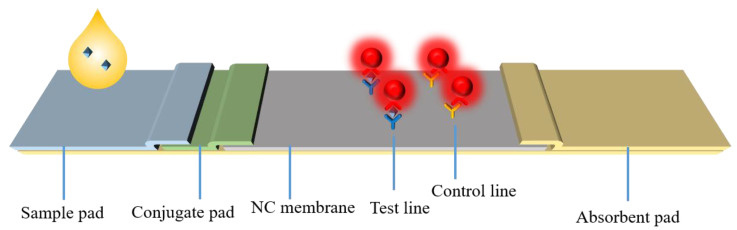
Schematic diagram of immunochromatographic diagnostic strip.

**Figure 2 membranes-13-00091-f002:**
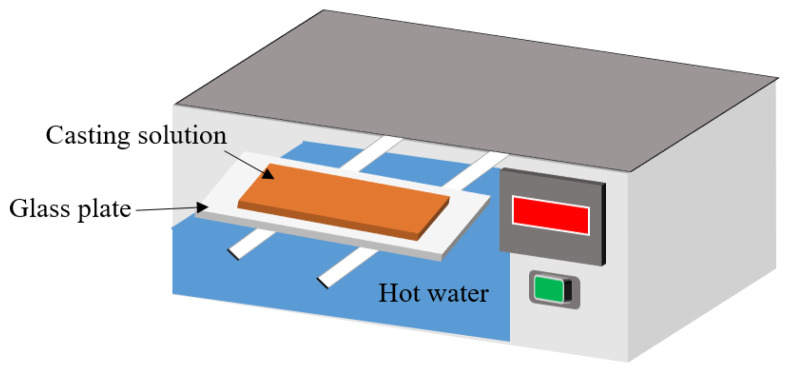
Schematic diagram of PVDF membrane preparation.

**Figure 3 membranes-13-00091-f003:**
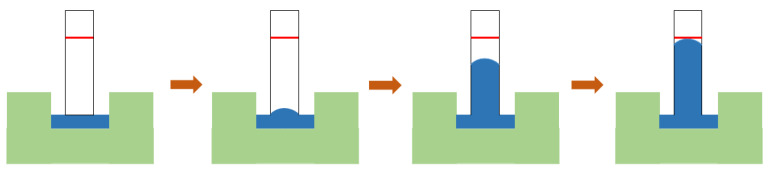
Diagram of capillary flow rate test.

**Figure 4 membranes-13-00091-f004:**
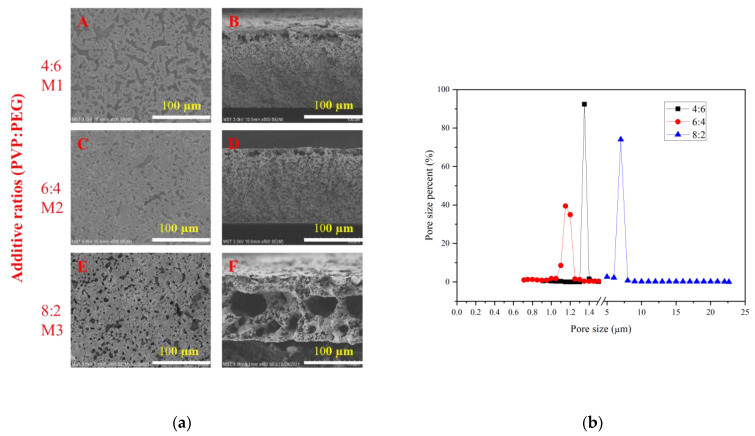
(**a**) SEM images of PVDF membranes prepared with different additive ratios of (**A**,**B**) 4:6, (**C**,**D**) 6:4 and (**E**,**F**) 8:2. (**b**) Pore size distribution of PVDF membranes prepared with different additive ratios.

**Figure 5 membranes-13-00091-f005:**
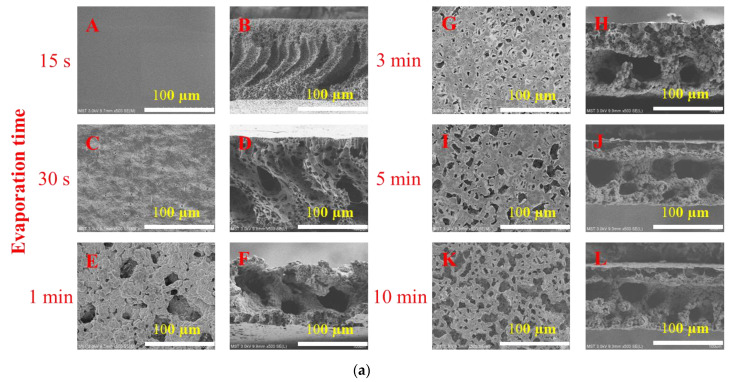
(**a**) SEM images of PVDF membranes prepared with different exposure time of (**A**,**B**) 15 s, (**C**,**D**) 30 s, (**E**,**F**) 1 min, (**G**,**H**) 3 min, (**I**,**J**) 5 min and (**K**,**L**) 10 min. (**b**) Pore size distribution of PVDF membranes prepared with different exposure times.

**Figure 6 membranes-13-00091-f006:**
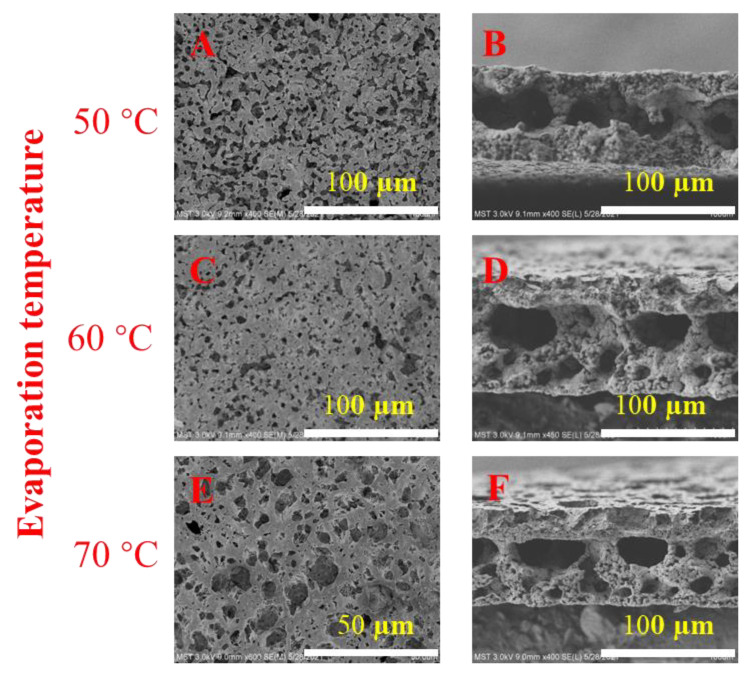
SEM images of PVDF membranes prepared at different vapor temperatures. (**A**,**B**) 50 °C; (**C**,**D**) 60 °C and (**E**,**F**) 70 °C.

**Figure 7 membranes-13-00091-f007:**
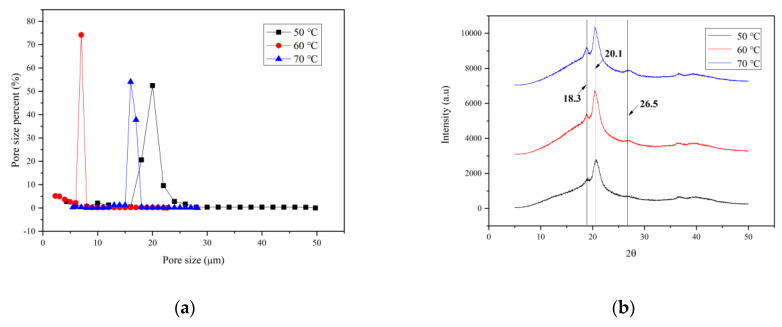
(**a**) Pore size distribution, (**b**) XRD spectra, (**c**) FT-IR spectrum, (**d**) zeta potential, (**e**) capillary flow rate and porosity, (**f**) tensile strength and elongation at break and (**g**) protein adsorption capacity of PVDF membranes prepared at different vapor temperatures.

**Figure 8 membranes-13-00091-f008:**
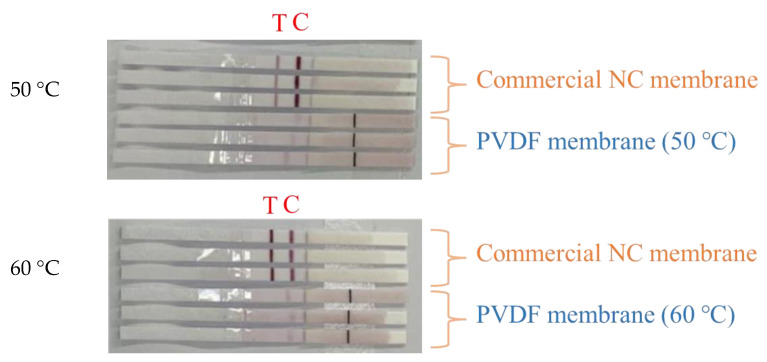
Application of colloidal gold test strips of PVDF membrane prepared at different vapor temperatures.

**Table 1 membranes-13-00091-t001:** Membrane codes for different additive ratios, exposure times and vapor temperatures.

Membrane Code	Additive Ratio (PVP:PEG)	Exposure Time (min)	Vapor Temperature (°C)
M1	4:6	5	60
M2	6:4	5	60
M3	8:2	5	60
M4	8:2	0.25	60
M5	8:2	0.5	60
M6	8:2	1	60
M7	8:2	3	60
M8	8:2	5	60
M9	8:2	10	60
M10	8:2	5	50
M11	8:2	5	60
M12	8:2	5	70

**Table 2 membranes-13-00091-t002:** Viscosity of casting solution with different additive ratios.

Additive Ratio (PVP:PEG)	Viscosity (cp)
4:6	7230 ± 42
6:4	12,660 ± 85
8:2	21,930 ± 127

## Data Availability

The data presented in this study are available on request from the corresponding author.

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
