# Peer review of "Preparation of Lateral Flow PVDF Membrane via Combined Vapor- and Non-Solvent-Induced Phase Separation (V-NIPS)"

_membranes, 2023, doi:10.3390/membranes13010091_

Round 1

Reviewer 1 Report

This work investigated the effect of additive as well as vapor temperature and exposure time during phase inversion of membrane fabrication on the membrane morphology and pore size. The application of the synthesized membranes as colloidal gold test strips was also demonstrated. Though this work showcases the new application for porous PVDF membrane and has the potential to draw attention from readers, the manuscript still has too many flaws to be published in the current version. The major concerns are as follows.

(1)   Lack of rationale and novelty in the membrane fabrication

-        A big part of this work is to develop porous membranes by employing additive and adjusting phase inversion conditions; however, its rationale behind additive selection has not been mentioned. The authors should fully search prior works related to the fabrication of porous membranes and review them properly in the introduction.

-        The addition of pore formation additives is not new for membrane fabrication via a phase inversion, the authors should provide proper literature to the reader.

-        Using PVP and PEG as additives to tune the porosity or hydrophilicity of the membrane is well established. The authors should provide the rationale or the purpose of each additive added to the system in the phase inversion.

(2)   Flaws in experimental, result, and discussion

-        The molecular weight of PVDF should be provided.

-        Please also provide the material specification and supplier for the model protein.

-        The concentration of polymer (PVDF + additives) in DMAC must be provided. As well, the total amount of additive in PVDF (in weight or volume or molar ratios or percentage) has to be mentioned in the membrane preparation section.

-        Before exposing the cast film to water vapor, was the chamber saturated with water vapor?

-        The viscosity of the polymer solution was discussed extensively as a key factor, influencing the pore formation of the membrane and thus must be provided.

-        It is doubted that PVP with high molecular weight was eluted, forming big pores. Normally, without solvent etching, PVP becomes entangled with the polymer matrix and remains in the structure of the membrane.

-        On page 5, please further explain how hydrophilicity significantly improves the thermodynamics of phase separation.

-        Pore formation and properties of the resultant membrane were mainly explained regarding the effect of PVP, how about the function of PEG?

-        In sections 3.2 and 3.3, the author described the pore formation based mainly on the effect of PVP and completely ignored the presence of PEG.

-        It seems that when increasing exposure time (> 3 min) the phase separation mechanism is dominated by VIPS. Please clarify and consider rewriting the content on page 6 lines 165-167.

-        In the abstract and conclusion, it is stated that the prepared PVDF membranes are suitable and effective for colloidal gold tests. However, the results on page 9 showed that the membranes failed to develop color for the T line and couldn’t form stable binding with the antibodies immobilized.

(3)   Other issues

-        Poor resolution for Figure 4 (a) and Figure 5 (a).

-        Unreadable axis in Figure 7 e – g. 

Author Response

Please see the attachment, including our responses.

Reviewer 2 Report

The article entitled “Preparation of lateral flow PVDF membrane via combined vapor and non-solvent induced phase separation (V-NIPS)” presents the detailed information about the synthesis and characterization of novel PVDF membranes for bio-assay methods.  The article is well written, however minor revision is needed before this paper can be considered for the publication.

1.      Authors can discuss the vapor and non-solvent induced phase separation (V-NIPS) in the introduction sections.

2.                  Authors can improve the resolution of SEM morphology of Fig. 4 and 5.

3.                  Authors can explain the reason for the selection of gravimetric method for the determination of pore size. What is the error percentage?

4.      The wettability of V-NIPS can be discussed.

5.      Authors can explain the reason for the selection of BSA for gold test strips.

6.      In the conclusion part, the authors could explain about future directions of these lateral flow PVDF membranes.

Author Response

(The authors gave the same response as above.)

Reviewer 3 Report

In an overall, it is a well-written paper and the experimental results are adequate. The results should be useful to the community of readers of Membranes. However, I believe that the paper requires a moderate revision (detailed below), before being accepted for the journal.

1)                 Please ensure all abbreviations are clearly defined when first mentioned. There are many abbreviations used in the abstract, which are unclear.

2)                 Please include research gap and key quantitative findings in the abstract with future implication of the study.

3)       The whole manuscript needs to be revised to reduce the percentage similarity. Currently, it has a high similarity index of 38% (excluding bibliography) when checked with Turnitin software.

4)           The literature review should be further enhanced with more recent and relevant publications to support the novelty of the work.

5)                 Line 179: There seems to be an error with the unit of exposure time.

6)             The authors reported that the effect of exposure time on the pore size of the membrane is unclear. Any similar report from literature or scientific justification to support this observation?

7)               Resolution of the SEM images, e.g., Figure 4a and Figure 5a, needs to be revised to enhance the clarity.

Author Response

(The authors gave the same response as above.)

Round 2

Reviewer 1 Report

The quality of the revised manuscript is significantly improved. However, the explanation for some points remains unclear and requires further clarification. These are listed below.

Point 1: Please provide the purpose or hypothesis of this work, regarding the use of the additive mixture. The following works might be helpful for this manuscript,

https://doi.org/10.1016/j.memsci.2013.04.009

https://doi.org/10.1016/S0011-9164(03)90081-6

Point 4 (also related to point 9): If the chamber is not saturated with water vapor before placing the cast film, when the cast film was brought in for such a short period of time (sort exposure), no water vapor might be generated. Each experiment might don’t have a constant vapor content. How to be sure that the difference in membrane structure was due to the exposure time, not vapor concentration?

Point 7: The explanation regarding the influence of hydrophilicity on the thermodynamics of phase separation is still unclear. When adding a hydrophilic additive, how it affects the miscibility of each phase or the shift in the binodal curve (the thermodynamics effect)?
